# A nanopore interface for higher bandwidth DNA computing

**Karen Zhang** [1,5], **Yuan-Jyue Chen** [2,5], **Delaney Wilde** [1], **Kathryn Doroschak**[1], **Karin Strauss** [2], **Luis Ceze** [1], **Georg Seelig** [1,3,4] **& Jeff Nivala** [1,4] ✉

DNA has emerged as a powerful substrate for programming information processing machines at the nanoscale. Among the DNA computing primitives used today, DNA strand displacement (DSD) is arguably the most popular, with DSD-based circuit applications ranging from disease diagnostics to molecular artificial neural networks. The outputs of DSD circuits are generally read using fluorescence spectroscopy. However, due to the spectral overlap of typical small-molecule fluorescent reporters, the number of unique outputs that can be detected in parallel is limited, requiring complex optical setups or spatial isolation of reactions to make output bandwidths scalable. Here, we present a multiplexable sequencing-free readout method that enables real-time, kinetic measurement of DSD circuit activity through highly parallel, direct detection of barcoded output strands using nanopore sensor array technology (Oxford Nanopore Technologies' MinION device). These results increase DSD output bandwidth by an order of magnitude over what is currently feasible with fluorescence spectroscopy.

The predictability of Franklin-Watson-Crick base pairing has enabled the construction of a wide range of DNA-based computing systems, including amplifiers[1], Boolean logic gates[2–4], chemical reaction networks[5,6], oscillators[7], molecular diagnostics[8], and neural networks[4,9]. These circuits rely on a basic molecular primitive called toehold-mediated DNA strand displacement (DSD)[10,11]. DSD is a competitive hybridization reaction in which a single-stranded DNA (ssDNA) or RNA input displaces an incumbent output strand from a complementary binding partner. Multiple DSD reactions can cascade to create a complex, yet programmable reaction network. Due to the simplicity of this mechanism and its ability to operate in both cellular and enzyme-free settings, DSD circuits are widely applied, readily scaled up, and constitute some of the largest molecular circuits designed by humanity so far.

Readout of DNA strand displacement activity typically relies on a fluorophore-quencher strategy, in which a DSD reaction is designed to liberate a fluorophore-labeled strand from a quencher-labeled strand. The fluorescent properties of the reaction can then be measured with a spectrofluorometer to determine the relative concentration of the free fluorophore-labeled strand[12]. Several unique fluorophores can be combined in a single system to label different DSD components, however spectral overlap amongst fluorescent dyes (e.g. FAM, TAMRA, Cy5) ultimately limits this multiplexing to around 3 or 4 unique labels[13]. Even then, achieving this scale of orthogonal readouts requires complex and expensive instrumentation, such as multiple sets of optical filters[14–16]. As the field of DNA computing progresses, more scalable detection methods are critical for circuit multiplexing, multilayer kinetic characterization, and debugging[2]. Ideally, such readout technologies would also be inexpensive, portable, and fast, which would support applications such as rapid diagnostics.

Nanopore sensing is a simple single-molecule detection method applicable to a wide range of analytes, from small molecules and peptides to nucleic acids and proteins[17–21]. Nanopore array technology has recently been commercialized for DNA and RNA sequencing, in which the technology's key advantages compared to other sequencing technologies include instrument portability and real-time data

[1]Paul G. Allen School of Computer Science and Engineering, University of Washington, Seattle, WA, USA. [2]Microsoft Research, Redmond, WA, USA. [3]Department of Electrical and Computer Engineering, University of Washington, Seattle, WA, USA. [4]Molecular Engineering and Sciences Institute, University of Washington, Seattle, WA, USA. [5]These authors contributed equally: Karen Zhang, Yuan-Jyue Chen. ✉e-mail: jmdn@cs.washington.edu

streaming. Nanopore sequencing is facilitated by enzyme-assisted translocation of nucleic acid strands through the pore, making it possible to achieve single-nucleotide sequence resolution of long DNA fragments[22]. In this sequencing approach, a ligation step is performed prior to sequencing to attach the target strands to adapter DNA fragments bound to the requisite motor proteins. This procedure is currently incompatible for DSD circuit readout, especially when analysis of circuit kinetics is desired. Furthermore, strands in DSD circuits are usually too short (around 15–50 nt) to be reliably characterized via conventional nanopore sequencing.

Previous studies have explored non-sequencing-based nanopore sensing techniques for nucleic acid strand detection in the context of DNA computing[23]. For example, nanopore detection of unlabeled DNA and RNA circuit outputs has been demonstrated with micro-droplet systems, wherein the target strand is electrophoretically pulled through a protein pore connecting two droplets[24–26]. However, such studies have not yet shown these systems to be quantifiable nor multiplexable. Meanwhile, solid-state nanopores have been used to characterize the double-stranded outputs of DNA assembly reactions[27] and the presence/absence of streptavidin-based DSD output tags[28], but are also difficult to multiplex. Apart from DNA computing, nanopore technology has also been used for miRNA detection, facilitating disease diagnostics[23,29]. Further studies in this area demonstrated that peptide nucleic acid (PNA) and polyethylene glycol (PEG) probes are effective at targeting specific miRNAs for nanopore detection[30,31]. However, it also remains challenging to multiplex detection of multiple miRNAs using these probes.

Here, we developed a multiplexable reporting strategy that utilizes off-the-shelf nanopore sensing array technology (Oxford Nanopore's MinION)[22] to dynamically monitor many DSD circuit reactions in real-time, enabling more scalable readout of output kinetics using an inexpensive, portable device. First, we demonstrate successful kinetic characterization of a single DSD circuit operating within a nanopore array flow cell with results comparable to traditional fluorescence-based reporting systems. We then design and characterize dozens of nanopore barcodes that can be used for multiplexed DSD circuit reporting. In this strategy, barcodes are classified directly from raw nanopore signal data using machine learning. Finally, we implement this reporting strategy to discriminate amongst combinations of three single nucleotide variants (SNVs) of the let-7 microRNA family.

## Results and discussion

### Detection of immobilized ssDNA using a MinION

To develop our reporting method, we first designed a modified DNA strand architecture that would allow us to sense ssDNA DSD outputs using a nanopore (Fig. 1a). We did this by modifying the output strand to include a 3′ biotin modification. We reasoned that, when conjugated to streptavidin, this modification would prevent displaced ssDNA from fully translocating through the nanopore after its free 5′ end is electrophoretically captured. The nanopore ionic current signal would then be dependent on the sequence of the strand immobilized within the pore[18,32,33]. To verify this approach, we tested biotin-modified DNA strands in the presence of streptavidin on a MinION nanopore sensor array using a custom run script that briefly reversed the applied voltage across the nanopore array every ten seconds, allowing us to repeatedly capture strands from the bulk solution and then eject them electrophoretically[34]. Repeatedly sampling strands from the bulk flow cell solution in this manner allowed us to detect DNA-specific nanopore capture events at a concentration-dependent frequency (Fig. 1b–d).

### Characterization of DSD circuit kinetics

After confirming that we could detect free biotin-streptavidin-modified ssDNA strands, we next tested whether this technique could be coupled to sense the free output strand concentration of a catalytic seesaw-based DSD reaction (Fig. 2a, b). Seesaw circuits have previously been used to build large-scale logic circuits and neural networks[2,4,9]. In this circuit architecture (Circuit 1 with Barcode A1, see Supplementary Tables 1 and 2), an input strand displaces a gate-bound output strand. A fuel strand then binds to the gate and displaces the input, freeing it to trigger more of the output. As such, a seesaw gate can catalytically amplify its input, a critical step for restoring signals. A catalytic reaction can increase the overall displacement percentage up to 100%[2,35], and the comparison between a catalytic reaction and a non-catalytic reaction is shown in Supplementary Fig. 1. To allow detection of the seesaw gate output strand, we designed the strand with the requisite 3′ biotin modification, in addition to a short polyT linker between this modification and the rest of the sequence. We then hybridized the output strand with the complementary gate strand to form the gate complex. The gate complex bound with streptavidin was then added to a MinION flow cell in two different conditions: with and without the presence of an input strand. Results from these experiments are shown in Fig. 2c (raw traces in Supplementary Fig. 2). Specifically, we determined the output strand's average nanopore strand capture frequency in 5-minute intervals, allowing us to measure the reaction kinetics directly within the flow cell solution (Methods). In parallel, we also monitored an identical reaction using a traditional fluorophore-quencher-based reporter gate and fluorometer (Fig. 2b, c). We found that nanopore samples had comparable output kinetics to those measured using the fluorophore-quencher approach, indicating that our nanopore method can be used to accurately monitor the reaction kinetics of a catalytic DSD reaction. We also noted that samples with no input added showed higher levels of output strand leak when using the fluorescent reporting method. We hypothesized that this circuit leakage was caused by the interaction between the seesaw gate and the fluorescent reporter gate[2] (Fig. 2b). Because our nanopore readout strategy detects the ssDNA output strand directly, it does not require an additional reporter gate, thus the reporter leak was not observed on the nanopore-based kinetics plot. To verify this hypothesis, we tested a clamped seesaw gate (Supplementary Table 3) that has previously been shown to suppress reporter leakage[2]. Indeed, the no input sample kinetics using a clamped seesaw gate more closely matched the no input nanopore readout (Supplementary Fig. 3). We also confirmed that the circuit input strands, fuel strands, and intermediate complexes are not extracted as captures by our analysis pipeline, and thus do not contribute to our measured output concentration (Supplementary Figs. 4 and 5). In addition, we note that the gate complex contained a 15 nt overhang to make it compatible with the fluorescence reporter gate, and that this overhang increases the background capture rate of the output strand in the pore but is accounted for by normalizing against background (i.e. no input conditions) (see Methods). Capture of the gate complex can be substantially reduced by removing the D6 overhang domain from the output strand shown in Fig. 2a (Supplementary Fig. 6).

### Barcoded output strands for circuit multiplexing

Having developed a technique to measure DSD kinetics of a single reaction using a nanopore array, we next sought to explore multiplexed detection through barcoding of the output strand. We started by determining the nanopore-addressable region of the output strand. To do this, we designed and tested a series of ssDNA output strands that contained a single nucleotide variation at a different position within a nine-nucleotide window directly upstream of the biotinylated 3′ end. We found that mutations within a six-nucleotide subset of that window manifested substantial changes to the mean ionic current signal relative to the original output strand (Fig. 3a and Supplementary Fig. 7). We designated this sensitive sequence region as the output strand's nanopore-addressable barcode.

As an initial test to see how accurately barcodes of dissimilar sequences could be discriminated amongst each other, we randomly

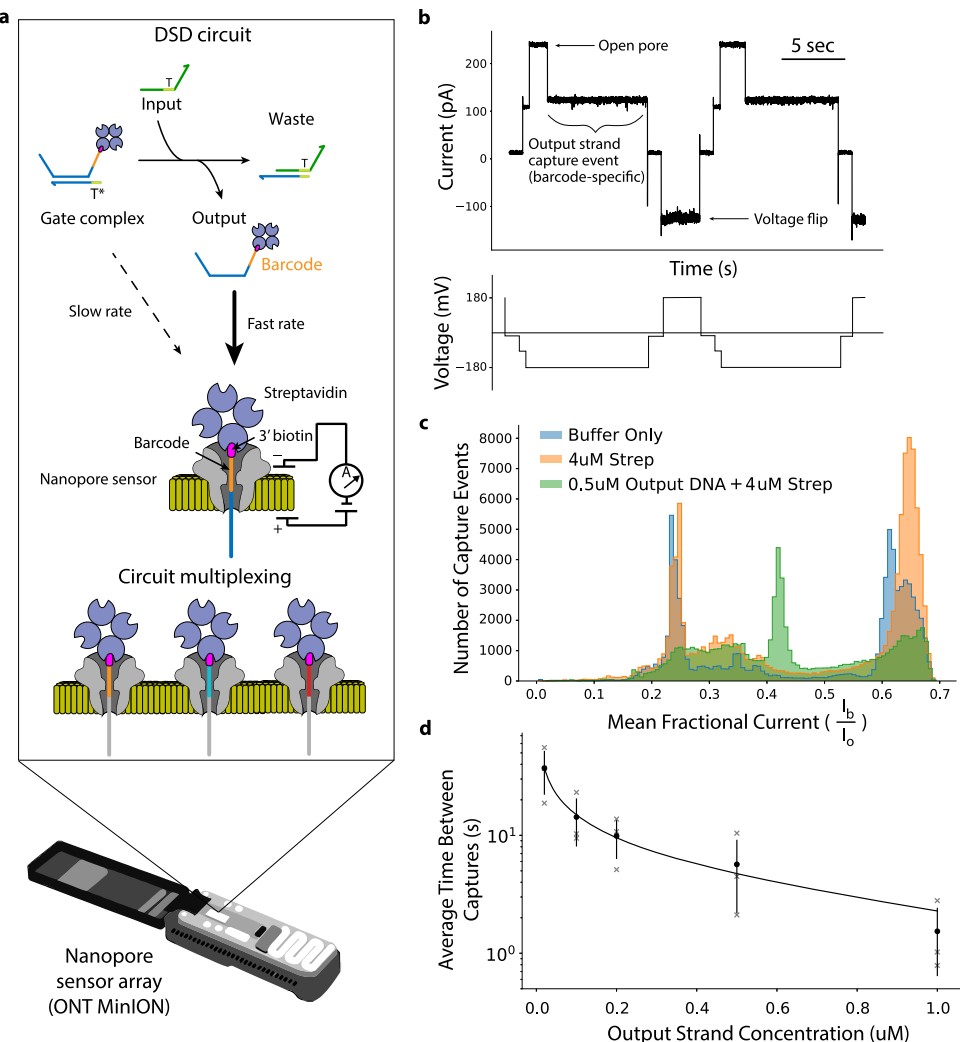

**Fig. 1 | Nanopore detection of DNA circuit outputs. a** DSD circuit detection with a nanopore sensor. Circuit components are mixed and loaded onto a nanopore sensor array for real-time readout. Input strands react with the gate complex displacing the barcoded and 3' labeled biotin-streptavidin output ssDNA, which is then free to be captured and read by a nanopore sensor. The nanopore sensor array is capable of distinguishing different output strand barcodes, enabling circuit multiplexing. **b** Example raw nanopore data showing repeated capture and ejection of biotinylated ssDNA output strands. DNA capture events manifest ionic current

drops from open-pore to a lower ionic current level. Strands are ejected from the pore by reversal of the applied voltage. **c** Histogram showing the distribution of mean fractional current from capture events belonging to three samples: nanopore running buffer, running buffer with 4 uM streptavidin, and running buffer with both 4 uM streptavidin and 0.5 uM 3' biotinylated ssDNA. **d** Standard curve showing the relationship between average time between output strand capture and strand concentration. Error bars represent ± standard deviation of three replicates, with each replicate using a different barcoded strand.

addressed a set of ten different output strands (Circuits 0-9 with Barcodes A1-A9, respectively, see Supplementary Tables 1 and 2, also Supplementary Fig. 8) and characterized them on a MinION. Capture event fractional mean currents for each of these barcodes showed that while some of the barcodes displayed distinct signal levels, indicating high separability based on this feature alone, some barcodes had significant overlap (Fig. 3b and Supplementary Fig. 9). Fitting a logistic regression model to these means yielded a single-molecule classification accuracy of ~32% on a withheld test set (Methods). Meanwhile, a Random Forest model trained on five features of the raw signal (mean, median, standard deviation, maximum, and minimum) yielded an accuracy of ~64% (Supplementary Fig. 10). To capture more signal information that could allow for better discrimination amongst the different barcodes, we used a 4-layer Convolutional Neural Network (CNN) to automate feature extraction and classification directly from the raw nanopore signal (Fig. 3c and Methods). CNN models trained on data collected for each of the random barcodes yielded a single-molecule classification accuracy of ~72% on a withheld test set (Fig. 3d and Methods). We then chose two barcodes from this set (barcodes A0

and A5) and tested them as output reporters for two orthogonal see-saw gates (Fig. 3e). We did this by mixing the two seesaw circuits (Circuits 0 and 5) in solution and then selectively triggering one or the other with the addition of their corresponding input strand, depending on the experimental condition, immediately prior to loading into a MinION flow cell. We then quantified each circuit's reaction kinetics over the course of the experiment using the CNN to demultiplex the output strand capture signals. Results from these experiments showed that only the selectively triggered circuit barcode in each of the two experimental conditions had an elevated output signal, indicating that our classifier was able to correctly identify the barcoded circuits that were present and activated against a background of barcodes that were unactivated or not present in the sample (Fig. 3e and Supplementary Fig. 11).

To further develop the multiplexing potential of this method, we next tested two sets of rationally-designed barcodes (Sets B and C, see Supplementary Table 2 and Supplementary Fig. 8). Set B was based on a predictive nanopore signal model of DNA kmers[36], which we used to select a set of thirteen 6-kmer barcode sequences predicted to have

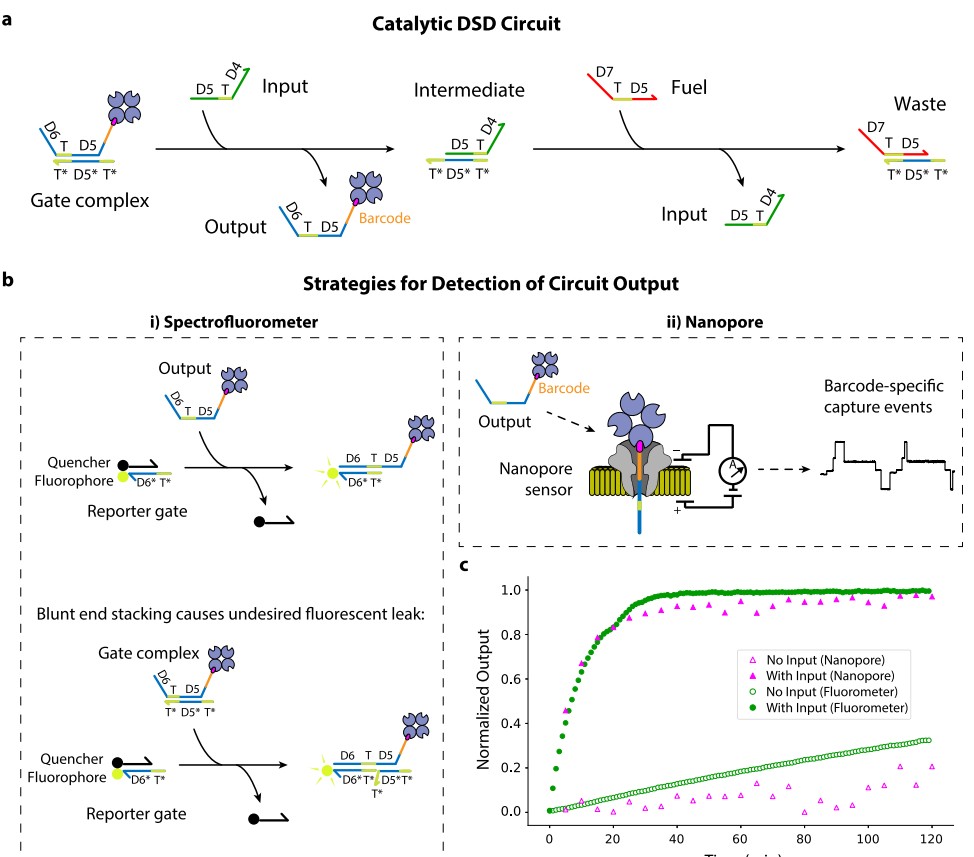

**Fig. 2 | Nanopore and fluorometer-based comparison of catalytic DSD circuit kinetics. a** Diagram of a seesaw catalytic DSD circuit. Single-stranded input displaces the 3′ labeled biotin-streptavidin single-stranded output from the gate complex, forming an intermediate with the bottom strand. The displaced output strand is now free to capture in a nanopore. Single-stranded fuel displaces input from the intermediate, recycling the input. **b** Displaced output ssDNA can be read out using two detection strategies: (i) *Spectrofluorometer-based detection*. Output strand displaces the quencher-labeled strand from a fluorophore-labeled reporter gate complex, triggering fluorescence. Blunt end stacking can occur in this reporter strategy, in which a double-stranded gate complex displaces the quencher strand, resulting in undesired leaky fluorescence. (ii) *Nanopore-based detection*. A single-stranded output strand is displaced by an input strand and can then be captured in a nanopore, resulting in a detectable drop in ionic current that is diagnostic of the strand's barcode sequence. **c** DSD reaction kinetics plot determined by nanopore (pink) and, for comparison, a spectrofluorometer (green), showing the normalized concentration of output strand after addition of input.

nearly non-overlapping ionic current levels (Supplementary Fig. 12). Set C included the use of abasic sites within the barcode sequence. We reasoned that abasic sites, which lack a nitrogenous base, would yield higher ionic current levels than barcodes composed entirely of standard bases, stretching the dynamic range of barcode signal space. In total, we characterized 36 different barcodes, which we were able to classify amongst each other with a non-trivial single molecule accuracy of ~60% using the 4-layer CNN. We additionally benchmarked barcode classification accuracies using ResNet-18, a more powerful CNN popular in computer vision applications[37]. The ResNet-18 model yielded improved 36-way classification accuracies of ~93% and ~67%, with and without the use of a classifier confidence filter (Fig. 4a, Supplementary Fig. 13 and Methods). From this collection, we identified a subset of ten barcodes with the most separable signal levels (Fig. 4b and Supplementary Fig. 14) and achieved an average single-molecule classification accuracy of ~96-97% after training and testing the ResNet-18 CNN on this limited set (Fig. 4c and Supplementary Fig. 13). We then performed another set of multiplexed seesaw circuit experiments using this subset of optimized barcodes and the corresponding CNN classifier. In these experiments, up to three circuits were activated at a time in samples containing a total of five orthogonal seesaw circuits. In contrast to our previous nanopore experiments, input strands were introduced into the samples, and the reactions were allowed to reach a steady state prior to analysis on the nanopore (Methods). Specifically, our classifier was tasked with distinguishing amongst all ten barcodes

in our optimized set, although only five of those (barcodes B10, B7, C13, C8, and C12) were physically present in the samples, and either two (barcodes B7 and C12) or three (barcodes B10, C13, and C8) of these circuits were activated with their corresponding input strands, depending on the experimental conditions. Results from these experiments are shown in Fig. 4d (raw traces in Supplementary Fig. 15). We found that circuits with barcodes that were activated in each sample showed elevated capture levels compared to all other barcodes. These results demonstrate the ability of our nanopore reporting strategy to multiplex DSD circuit readouts beyond what is possible with current fluorescence-based technologies.

## Multiplexed detection of let-7 miRNA SNVs

Finally, we sought to test our reporter system in a multiplexed diagnostic application by discriminating amongst combinations of single nucleotide variants (SNVs) of the let-7 microRNA (miRNA) family. The let-7 family of miRNAs plays an important role in human development and can be used as biomarkers for disorders such as cancer[38]. Inspired by Chen et al.[39], we designed competitive three-stranded probes for the detection of three variants: let-7a, let-7e, and a truncated version of let-7c (Fig. 5a, b, also Supplementary Table 4). Let-7c was truncated to reduce potential probe crosstalk by destabilizing an undesired wobble base pairing between let-7c miRNA and let-7a probe. We chose three barcodes from our designed set and assigned a different one to each variant's probe, then trained a three-way ResNet-18-based classifier to

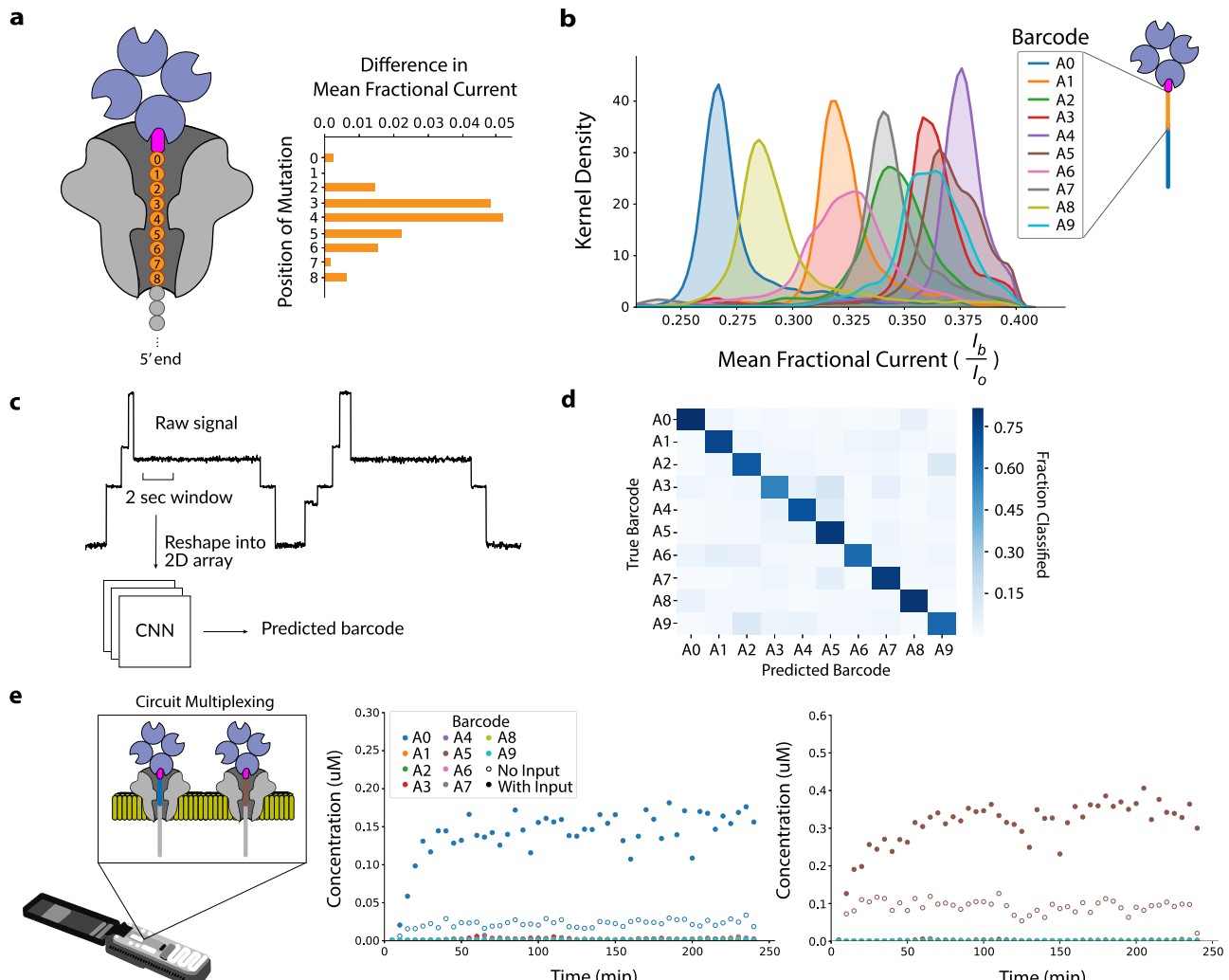

**Fig. 3 | Initial barcode design, classification, and multiplexing. a** Mapping the output strand nanopore-sensitive region. The plot shows the absolute change in mean fractional ionic current manifested by a single-nucleotide mutation at different positions along the strand. **b** Density plot showing the distribution of mean fractional current for each barcode in Set A. Each distribution is composed of ~14500 data points. **c** To perform classification, a two-second (20,000 data point) window of the output strand capture event signal is extracted, reshaped into a 2D array, and then used as input to a 2D ResNet-18 CNN. The CNN's output is a barcode prediction. **d** Confusion matrix showing classification results of CNN inference on a barcode test set, which achieved an average single-molecule accuracy of 72%. **e** DSD reaction kinetics plot of two different circuits (barcoded with A0 or A5) multiplexed on the nanopore device. Each gate complex was present at 0.5 uM, fuel strand at 2 uM, input strand at 0.2 uM, and streptavidin at 2 uM. Two samples were prepared: No Input and With Input. The left plot shows the detection of Barcode A0 output for both samples. The right plot shows the detection of Barcode A5 output for both samples.

discriminate amongst these barcodes. If a particular miRNA variant is present, its corresponding probe should be triggered, allowing its barcoded output strand to be detected by the nanopore sensor array. To avoid amplifying crosstalk leakage, we did not use fuel strands for our probes. We tested this strategy with experiments containing all three probes in a single reaction solution, which we then challenged with different combinations of let-7a, let-7c, and let-7e RNA inputs prior to measurement with a MinION. Results from these experiments showed clear enrichment of barcodes corresponding to the probes that were triggered by their cognate RNA input (Fig. 5c and Supplementary Fig. 16). Although mostly orthogonal, RNA/probe crosstalk was observed in the case where the addition of let-7e input triggers the release of let-7a probe output slightly above background levels. This is likely due to the relatively large distance between the let-7e SNV and toehold-complementary region (T1*), which lowers the kinetic penalty of let-7e inputs hybridizing to let-7a probes. These results are also consistent with fluorescent reporter-based measurements, indicating that this crosstalk is not a result of our nanopore-based reporters (Supplementary Fig. 17). Overall, these

results demonstrate successful multiplexing of a microRNA-based diagnostic.

In summary, we have developed a reporter strategy for DSD reactions using nanopore sensing. Our system holds key advantages over fluorescence-based methods, including greater multiplexing and real-time readout using an inexpensive, portable device with flow cells that can be re-used for multiple analytical samples (Supplementary Fig. 18). Further, compared to fluorophore-quencher pair-based reporter systems, our method uses DNA sequence-based barcodes (not small molecules) and so only requires a single type of DNA modification (biotinylation), which is simpler to synthesize and is not susceptible to photobleaching. However, possible limitations of our nanopore interface include a lower sensitivity compared to fluorometers, lower time resolution, and the inability to multiplex samples under different conditions due to the lack of physical barriers to separate samples in a flow cell. Another area for potential improvement is classifier prediction accuracy, which can be influenced by classifier type and the number of training examples (Supplementary Fig. 19) or by imposing a confidence threshold (Supplementary Fig. 13).

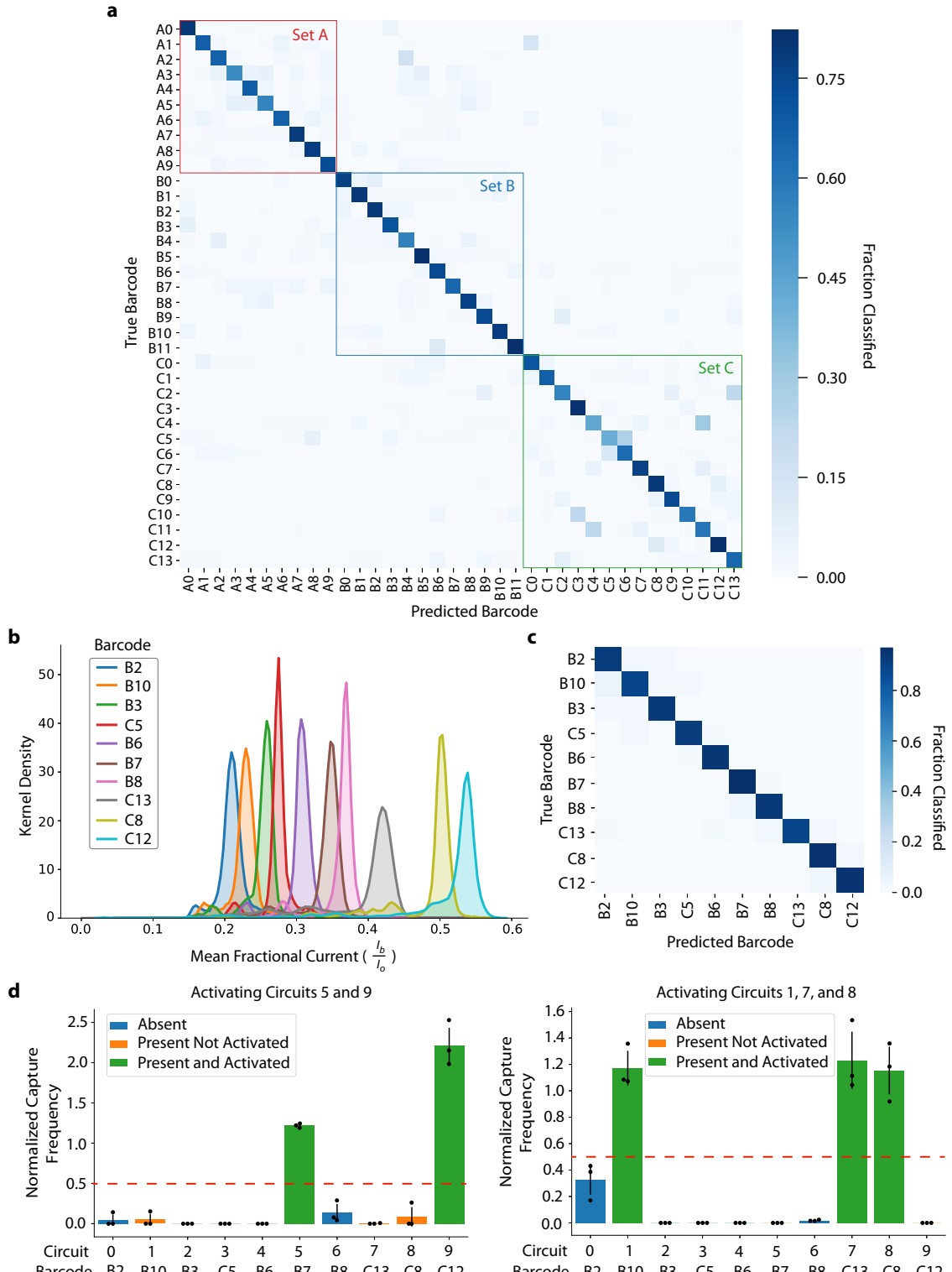

**Fig. 4 | Developing an optimized nanopore barcode set. a** Confusion matrix showing results of ResNet-18 CNN trained on all 36 characterized barcodes. CNN achieved an accuracy of ~67% on the test set. **b** Density plot showing the distribution of mean fractional current for each barcode selected for the highly separable set. Each distribution is composed of ~13000 data points. **c** Confusion matrix showing results of ResNet-18 CNN trained only on the selected highly separable barcodes, which achieved an accuracy of 93% on the test set. **d** Bar plots comparing normalized capture frequencies of each circuit output in two multiplexed samples. Circuits 1, 5, 7, 8, and 9 (with Barcodes B10, B7, C13, C8, and C12, respectively, see Supplementary Tables 1 and 2) were present in each sample. In the left plot, inputs for Circuits 0 and 5 were added. In the right plot, inputs for Circuits 1, 7, and 8 were added. Each gate complex was present at 0.2 uM, input at 0.2 uM, and streptavidin at 3.2 uM. Error bars represent ± standard deviation of three biological replicates.

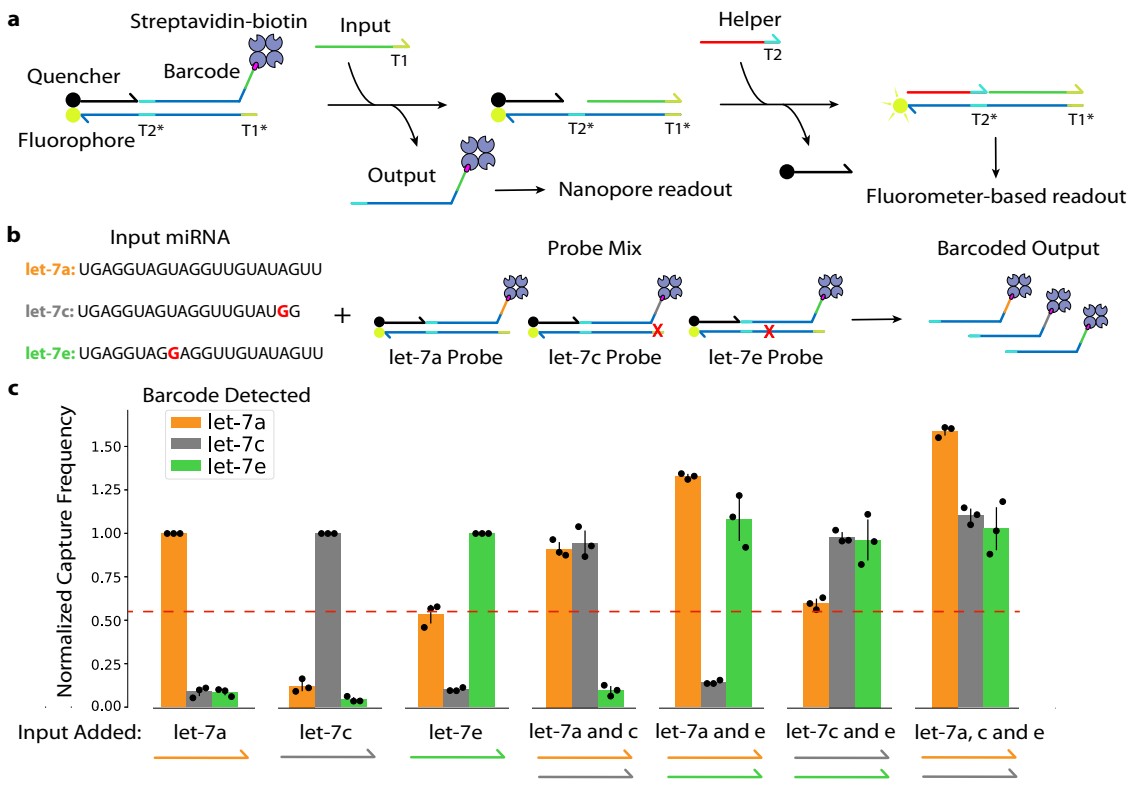

**Fig. 5 | Multiplex detection of let-7 microRNA SNV variants. a** Two-step probe mechanism. miRNA input hybridizes to the first toehold region (T1) on the bottom strand, displacing the barcoded output strand for nanopore readout. Next, a helper strand hybridizes to the second toehold region (T2) on the bottom strand, displacing the quencher strand and enabling spectrofluorometer-based readout. **b** Multiplex experiment setup. Input strands are synthetic let-7 miRNA SNV variants (sequences shown with SNVs bolded in red) which are added to a mix of probes. A red X denotes the location of bases complementary to SNVs. Hybridization of an input with the correct probe releases its respective barcoded output for nanopore detection. **c** Seven samples, each containing three let-7 probes, were prepared. A different combination of input strands was added to each sample, as visualized under each cluster of bars. The bar plot shows the capture frequency of each barcoded output in each sample as measured on the nanopore after reaching a steady state. Error bars represent ± standard deviation of three biological replicates.

Future work will be aimed at: 1) further expansion of the barcode space, for example, with chemical modifications to the DNA that could expand the dynamic range of barcode signal space[40], and/or the ability to read sequential barcode regions within a single output strand[25,41,42]; and 2) increasing the reaction speed and sensitivity of DSD reactions, for example by spatially-localizing DNA components to the nanopore sensor membrane[43,44]. Increasing the scale and speed of our detection strategy for more complex molecular computing architectures, such as cascaded circuits or oscillators, will further take advantage of our method's ability to generate both multiplexed and kinetic readouts. These advancements would expand the capabilities of molecular computing tools by facilitating the design and facile characterization of more complex circuits, bringing forward new opportunities for their application in medical diagnostics[45], therapeutics[46], biomolecular-based instruments[47], and molecular information processing[4,8].

## Methods
### MinION experiments
All MinION runs were performed using R9.4.1 flow cells (Oxford Nanopore Technologies). All runs were configured at a temperature of 30°C, bias voltage of −180 mV, sampling frequency of 10 kHz, and static flip frequency of 15 sec with the use of a custom MinKNOW script (available from ONT) which allows for the collection of raw current data. All samples were suspended in 1X C17 buffer (2 M KCl, 100 mM HEPES, pH 8). Samples were pipetted into the flow cell priming port. If running multiple samples on the same flow cell, the flow cell was washed with 3 mL 1X C17 for 5 min between samples. When not in use, flow cells were stored at 4°C in C18 buffer (150 mM potassium

ferrocyanide, 150 mM potassium ferricyanide, 25 mM potassium phosphate, pH 8).

### DSD circuit preparation
All DSD circuit components were ordered from Integrated DNA Technologies (IDT). All components were stored at −20°C in the long term and 4°C in the short term (less than a month). Circuit gate complexes were constructed by mixing 4 nmol output strand with 4 nmol bottom strand in 0.8X C17. Strands were annealed in a thermocycler starting at 95 °C for 2 min and decreasing 1°C every 1 min cycle for 75 cycles. New England Biolabs (NEB) Purple Gel Loading Dye (no SDS) was added to the annealed product at a final concentration of 1X. The double-stranded annealed product was then separated from excess ssDNA on a 10% non-denaturing polyacrylamide gel. Bands corresponding to annealed product were cut and submerged in 1X C17 for at least 24 h to elute.

Fluorescent reporters were prepared by mixing quencher strands at 30% higher concentration than fluorophore strands in 1X C17. Strands were annealed using the thermocycler protocol described above.

### Titration experiments
The standard curve is an average of titration data from three different output strands. In each titration experiment, the output strand was run on the MinION at concentrations of 0.02, 0.1, 0.2, 0.5, and 1 uM. Each sample also contained streptavidin (NEB #N7021S) at 4 uM and was suspended in 1X C17. To ensure enough captures were collected for data analysis at low concentrations, the 0.02 uM sample was run for

20 min, the 0.1 uM sample for 15 min, and the rest of the samples for 10 min.

## Kinetics analysis of a single DSD circuit

The sample for this experiment consisted of 0.5 uM gate complex, 2 uM streptavidin, and 2 uM fuel strand suspended in 1X C17 to a total volume of 200 uL. Immediately prior to loading into the MinION flow cell, 0.2 uM input strand was added to initiate the reaction. Ionic current data was collected over the course of four hours on the MinION. The average time between captures (TBC) was calculated for each five-minute interval throughout the run and normalized as follows, where $TBC_{background}$ is the average TBC of the first five data points in the No Input sample and $TBC_{saturated}$ is the average TBC of 0.5 uM free output strand measured on its own in a separate sample.

$$TBC_{normalized} = \frac{TBC - TBC_{background}}{TBC_{saturated} - TBC_{background}} \tag{1}$$

Fluorescence kinetics data of the same circuit at a total volume of 100 uL was collected every 1 min using a microplate reader (Synergy HTX, Multi-Mode reader, Biotek). Excitation (emission) wavelengths were 485 nm (528 nm) for dye FAM and 620 nm (680 nm) for dye Cy5. The sample composition was identical to the nanopore sample except for the addition of a fluorescent reporter gate at 3 uM.

## Multiplexing experiments

Samples for the two-circuit multiplexing experiment consisted of 0.5 uM gate complex and 2 uM fuel strand from each circuit, along with a total of 4 uM streptavidin, suspended in 1X C17 to a total volume of 200 uL. Immediately prior to loading into the MinION flow cell, 0.2 uM input for each circuit was added.

Samples for the five-circuit multiplexing experiment consisted of 0.2 uM gate complex from each circuit and a total of 3.2 uM streptavidin suspended in 1X C17 to a total volume of 200 uL. 0.4 uM of each desired input was added to the samples, which were then immediately placed in a 30°C incubator and allowed to react to steady state over the course of three hours (circuits were previously characterized on a fluorometer to ensure three hours is adequate time to reach steady state, which we define as the point when output fluorescence does not change over 5% for at least one hour). Samples were then loaded into a MinION flow cell for analysis. Each sample was run for 10 min.

## Let-7 probe preparation

All let-7 probe components were ordered from IDT. Helper strands and fuel strands were PAGE purified, while all other strands were HPLC purified. All components were stored at −20°C in the long term and 4°C in the short term (less than a month). Probes were constructed by mixing top and output strands, each at 20% higher concentration than bottom strands in 1X C17. Strands were annealed in a thermocycler starting at 95°C and decreasing 1°C every 1 min cycle for 75 cycles. Glycerol was added to annealed products at a final concentration of 10% by volume. The annealed complex was then separated from excess ssDNA on a 10% non-denaturing polyacrylamide gel. Bands corresponding to the annealed product were cut and submerged in 1X C17 for at least 24 h to elute.

## Let-7 detection experiments

Samples for multiplexed detection of let-7 SNVs consisted of 100 nM of each let-7 probe, 130 nM of each let-7 helper strand, and 1200 nM streptavidin suspended in 1X C17 to a total volume of 200 uL. 50 nM of each desired let-7 microRNA was added to the samples, which were then immediately placed in a 30°C incubator and allowed to react to steady state over the course of one hour. Samples were then loaded into a MinION flow cell for analysis. Each sample was run for 10 min.

Fluorescence kinetics data of the same circuit at a total volume of 100 uL were collected every 1 min using a microplate reader (Synergy HTX, Multi-Mode reader, Biotek). Excitation (emission) wavelengths were 485 nm (528 nm) for dye FAM and 620 nm (680 nm) for dye Cy5.

## Nanopore data analysis

The nanopore data analysis pipeline is adapted from Cardozo et al.[30] and begins by isolating capture events from raw nanopore current. A capture event occurs when the nanopore current drops to 70% or below of its open pore level for longer than one millisecond. The fractional current throughout each capture event is then calculated, which is defined as the current observed during the capture event ($I_b$) divided by the open pore current ($I_o$). To separate putative output strand capture events from background noise, each capture event is passed through a filter that checks whether five of its signal features (mean, median, minimum, maximum, standard deviation) are within the expected range. The signal filter parameters were determined empirically. Specifically, for every different barcode we tested, we plotted raw current distributions for each filter parameter (e.g. raw current distribution of signal mean) and designed our parameter values to encapsulate each barcodes' signal feature peak distribution. Capture events are also passed through a length filter which discards events that have a capture duration of less than two seconds.

Initial exploration of classification on Barcode Set A was performed using the LogisticRegression and RandomForestClassifier models from scikit-learn. Each model was fitted to at least 14000 examples per class using a 80/20 train/test split. Classification accuracy is defined as the number of correct predictions (true positives + true negatives) divided by total predictions, converted to a percentage.

All subsequent classification of capture events were performed by Convolutional Neural Networks (CNNs) via PyTorch. For each capture event, the first two-second window of its raw nanopore signal is reshaped into a 2D array before being fed into the CNN. The 4-layer CNN comprises four convolutional layers, each with ReLU activation and max pooling, followed by a fully connected layer with log-sigmoid activation and an output layer containing a neuron for each barcode class. The ResNet-18 CNN model was obtained from the PyTorch torchvision.models subpackage. Reported accuracies and confusion matrices come from test results of the CNN after training with at least 3000 examples per class for the 36-way classifier and at least 20000 examples per class for the 10-way classifiers, using a 80/20 train/test split for 250 epochs. For analysis using confidence thresholds, the final output layer of the CNN was used as a measure of classification confidence for each barcode prediction, and predictions below a preset threshold (e.g. 0.9 or 90%) were filtered out.

Capture events from a particular class are quantified using a standard curve (determined from titration experiments) that relates the average time between capture events to sample concentration. The time between two capture events at a given pore is calculated by subtracting the end time of the first capture from the start time of the subsequent capture. In addition, any time periods between these capture events where the pore was occupied with noise or other barcode capture events (as determined by our filter and classifier) are also subtracted. The average time between captures is calculated from an aggregate list of total time between captures from all functional pores in a given experiment. Capture events can also be quantified using capture frequency, described in the next section.

## Capture frequency quantification

Capture frequency $f$ is defined as the number of reads for a given barcoded strand per good channel per minute. It is normalized using the following equation, where $f_{background}$ and $f_{saturated}$ represent the capture frequency of the barcoded strand when its circuit is not activated (no input) and when it is activated to a steady state, respectively

(these values were determined in separate experiments).

$$f_{\text{normalized}} = \frac{f - f_{\text{background}}}{f_{\text{saturated}} - f_{\text{background}}} \qquad (2)$$

### Statistics and reproducibility

The standard curve in Fig. 1d comprises three replicates, each with a differently barcoded strand and all successful. The circuit kinetics experiments in Figs. 2c and 3e were performed independently with no replicates. Three biological replicates were conducted for the five-circuit multiplexing (Fig. 4d) and let-7 variant detection (Fig. 5c) experiments with all replicates being successful.

For each of the density plots (Figs. 2b and 3b) and ML models trained, sample size was determined by the barcode class with the lowest number of captures obtained in nanopore experiments.

No further data were excluded from analyses. No randomization or blinding was used.

### Reporting summary

Further information on research design is available in the Nature Research Reporting Summary linked to this article.

### Data availability

A subset of the entire dataset (Fig. 4d, Replicate 1) analyzed in this study is publicly available (https://doi.org/10.5281/zenodo.6846356). The entire dataset, or specific subsets, can be obtained by request without any restrictions; it is prohibitively large (~600GB) to be made publicly available at this time. We will make every effort to send the dataset within a reasonable timeframe of 7 days.

### Code availability

The data analysis pipeline code used for extracting, filtering, classifying, and quantifying nanopore capture events can be found at https://github.com/uwmisl/dna-nanopore-computing (https://doi.org/10.5281/zenodo.6829445). Custom MinION MinKNOW run scripts can be obtained from Oxford Nanopore Technologies upon request.

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

## Acknowledgements
We thank members of the Molecular Information Systems Lab for helpful discussion and feedback on this work. We also thank Oxford Nanopore Technologies for providing the configurable MinION run script and discussions on its use. This work was supported in part by NSF EAGER Award 1841188 and NSF CCF Award 2006864 to LC and JN, NSF CCF Award 1954665 to GS and JN, and a sponsored research agreement from Oxford Nanopore Technologies.

## Author contributions
K.Z. and D.W. performed wet lab experiments. K.Z. and K.D. developed the data analysis pipeline and performed computational analyses. K.Z., Y.C., G.S., and J.N. designed experiments. Y.C., K.S., L.C., G.S., and J.N. supervised the project. J.N. conceived and directed the project. All authors contributed to the writing and editing of the manuscript.

## Competing interests
Y.C. and K.S. are employees of Microsoft. J.N. is a consultant to Oxford Nanopore Technologies. The remaining authors declare no competing interests.
