## [Peer Review File · Nature Communications]

Reviewers' Comments:

Reviewer #1:

Remarks to the Author:

DNA strand displacement (DSD) is a programmable gene material manipulation tool with wide applications in DNA nanotechnology, DNA computing, gene switch design, genetic detection. This paper reports an interesting strategy for simultaneous detection of multiple catalytic DSD circuits by sequencing-free discrimination of barcoded output strands released from the DSD reactions and expands the nanopore barcoded DSDs for multiple microRNAs detection.

However, the novelty of the work is lowered as both the nanopore DSD detection and nanopore DNA barcoding have been widely reported. For example, Kong et al (Chem. Commun., 2017, 53, 436-439) has demonstrated the capability of nanopore in detecting DSD by measuring the released output strand upon the target DNA binding. Further, various nanopore barcoding methods have been reported that detect DNA barcode sequences for multiplex biomolecular detection. In addition, the attachment of streptavidin to a DNA used in this report has already been extensively applied in detecting DNA sequence alteration, including nucleotide substitution and base modifications (PNAS 2009, 106, 7702-7707; PNAS 2012 109, 11504-11509; J Am Chem Soc 2010, 132, 17992-17995).

Also, the overall significance is reduced due to that the approach in this report is not proven to be accurate enough to discriminate a large number of barcode sequences required for multiplex detection, compared with reported DNA barcoding works (e.g., Molecular Cell, 2021; DOI: 10.1016/j.molcel.2021.07.006).

This current work employed Oxford Nanopore's sequencing device to facilitate DNA barcode detection. However, the accuracy for barcode sequence discrimination was only 32%, surprisingly lower than the reported >90% accuracy for the same setup. Although the Neural Network study indeed increases the 80/20 accuracy to 80%, it remains difficult to accurately discriminate a large scale of barcode sequences. This was why the authors finally selected only three barcode sequences containing different number of abases (0, 1, and 2) to validate the multiple DSD detection. But the nanopore has already been shown to detect the numbers of abase in a DNA sequence with much higher sensitivity and accuracy (e.g., Nat Nanotechnol 2010,5:798-806).

Another problem is DSD-based microRNA detections. Based on the experimental results, this DSD method does not demonstrate any superior or unique advantages over previously reported nanopore microRNA detection, as to the scale of the multiplex detection, low detection limit (sensitivity), detection of sequence-similar microRNAs and sensing in biofluid samples.

Overall, although the strategy for barcoded multiplex DSD detection is interesting, the results remain premature, cannot support the potential of this method in applications for large scale multiplex DSD screening and diagnostics.

Technical issues are as follow.

1. What is the function of the fuel strand in the catalytic DSD reaction? Is it to accelerate the DSD speed or to increase the overall displacement percentage up to 100%? The work should report the kinetics without the fuel strand. If the fuel strand is functional, it should be clarified why the fuel strand was not used to catalyze the DSD circuit for any microRNA detection? Or is the fuel strand has negative effect on the microRNA detection?

2. Fig.2a and the sequences in the tables show that after the first DSD, the duplex domain T*DS* in the gate complex is changed to DS*T* in the intermediate. The two duplexes are identical in the melting temperature, GC content, and length (19 bps), and free energy. So, can the released output re-bind the toehold on the gate strand (left T*) to reversely displace the input, if the displacement is driven by energetic favorite? Again, the kinetics to get the equilibrium displacement percentage without the fuel strand should be detected.

3. For the measurement of the DSD kinetics, the saturate frequency of the output strand has been used to normalize the output concentration. However, this parameter only shows the time-

dependent change but cannot show the actual DSD conversion efficiency (i.e. the percentage of the free output when getting the equilibrium. To calculate the DSD efficiency, it is strongly suggested that the frequency of free output strand alone (without any gate, input, and fuel strands) be used as the maximal frequency in place of the saturate frequency to normalize the output concentration.

4. It is very strange that in the mixture of various of DSD strands (gate, output, input, intermediate, fuel, the gate complex etc), only the output can be captured by the nanopore. It is also surprising that the gate complex with a 15-nt overhang cannot be captured by the nanopore, according to the report. Are these events not shown or not used to participate in data analysis? The authors may contact Oxford Nanopore to confirm these observations. It should be clarified under what conditions, the duplex with an overhang can or cannot be captured by the nanopore.

5. The report does not show any original nanopore traces that support each analysis result. The conclusion for each figure should be associated with original traces. In addition, Figures 2, 3 and 4 were for different experiments, but they repeatedly used the same nanopore recordings.

Reviewer #2:

Remarks to the Author:

Referee report

Re: NCOMMS-21-45691

A nanopore interface for high bandwidth DNA computing

by Karen Zhang, Yuan-Jyue Chen, Kathryn Doroschak, Karin Strauss, Luis Ceze, Georg Seelig, and Jeff Nivala

The work presented in this manuscript reports the use of nanopores for dynamically monitoring dynamically DNA strand displacement (DSD) circuit activity in real-time. This provides a fast, cheap, and scalable readout of kinetics. The barcodes obtained through the raw nanopore readout signals are directly classified using machine learning. The whole framework is evaluated for discriminating amongst combinations of three single nucleotide variants of the microRNA family. The main finding and novelty of this work relies on the fact that the kinetics monitored using the nanopore readout were shown to be comparable to those measured using a spectrofluorometer. Accordingly, it was shown that nanopores can be used as a real-time and cheap scheme to quite accurately monitor the reaction kinetics of a catalytic DSD reactions. This is a novel aspect in a field of intense research. Still some issues need to be clarified, as listed below, before the manuscript can be accepted for publication in Nature Communications.

Comments:

1. Fig.2c displays a comparison of using a nanopore and a spectrofluorometer. It would be insightful to discuss shortly the discrepancy in the kinetics in the 'no input' cases.

2. Could nanopore detection overcome fluorescent reporter-based detection in terms of accuracy? The authors claim that multiplexed detection is more efficient using nanopores. However, the discussion and the results shown in Fig.5 do not strongly support this claim. If indeed multiplexed detection is superior using nanopores, a stronger focus on this should be given in the manuscript.

3. Overall, the prediction discussed is not very high, especially compared with the accuracy of nanopores in detecting DNA.

The random barcodes yielded a single-molecule classification accuracy of about 72%. Did the authors attempt to optimise their network in order to enhance the accuracy or is this the highest accuracy obtainable? Is there a way to optimise the experimental setup and conditions instead of the Machine Learning part, in order to achieve a better prediction? Is there overall room for

improvement or is the accuracy of approximately 70% the upper bound? In that case, how would this compromise the use of nanopores as suggested here?

4. From the supporting information, it can be inferred that there was no development in the Machine Learning (ML) part. Were the tools used as is? Was the feature set tested, i.e. the type of features used? It has been shown that different feature sets in the nanopore signals can have a strong impact on the detection accuracy. In case, the authors did not consider this, they should at least comment on whether there is room for optimising the ML part, type of network, features, etc. in order to enhance the detection accuracy. That would further strengthen the use of nanopores over spectrofluorometers.

5. Regarding the experimental setup and conditions: Were other conditions, such as the molecules concentration etc, also tested? Were other circuits also used? If so, were the monitoring in those cases also successful?

6. The authors attempted to focus on the advantages of nanopores over spectrofluorometers. Are there, though, any limitations in the use of nanopores for a DSD kinetics readout? How barcode-specific is the framework proposed here?

7. How prone are the analysed results with respect to the filter that checks whether five of its signal features (mean, median, minimum, maximum, standard deviation) are within the expected range? Were the filters and the classifier optimised in advance to this study?

8. The training was performed with 3000 samples. The data set is not very large. Did the authors check, whether their results on the accuracy do converge or are prone to the sample size?

Minor Comments:

1. How is the steady state defined? In other words, is it certain that the steady state has been reached?

2. Typesetting error: on pg.9, line 5, do the authors mean 'All feature classification of capture events' instead of 'All future classification of capture events'?

3. Shortly discussing in the end the impact of the presented results and their applicability would add to the manuscript.

Point-by-Point Response to Reviewers
(A nanopore interface for higher bandwidth DNA computing)

We thank the reviewers for their careful consideration of our manuscript. We have now addressed or responded to all of the raised concerns, a summary of which follows here.

Reviewer #1:

...the novelty of the work is lowered as both the nanopore DSD detection and nanopore DNA barcoding have been widely reported. For example, Kong et al (Chem. Commun., 2017, 53, 436-439) has demonstrated the capability of nanopore in detecting DSD by measuring the released output strand upon the target DNA binding.

1.1 We appreciate the Reviewer for pointing out the work of Kong et al. In this work, authors detected SNVs by sensing the displacement of a streptavidin molecule from a probe using a solid-state nanopore. However, the detection scheme (presence/absence of streptavidin) and the requirement that this probe be attached to a large DNA carrier makes this strategy difficult to multiplex; in fact, no multiplexing was demonstrated in this paper. The large DNA carrier also consumes the input strand, and ultimately makes this readout system challenging to couple to the most widely used DSD circuit architectures that we used in our work. In any case, it is relevant to our work and we have now added the reference and an additional comment to Page 2, Lines 74-76:

“Meanwhile, solid-state nanopores have been used to detect the double-stranded outputs of DNA assembly reactions²⁷ and the presence/absence of streptavidin-based DSD output tags²⁸, but are also difficult to multiplex.”

...Further, various nanopore barcoding methods have been reported that detect DNA barcode sequences for multiplex biomolecular detection....Also, the overall significance is reduced due to that the approach in this report is not proven to be accurate enough to discriminate a large number of barcode sequences required for multiplex detection, compared with reported DNA barcoding works (e.g., Molecular Cell, 2021; DOI: 10.1016/j.molcel.2021.07.006).

....This current work employed Oxford Nanopore’s sequencing device to facilitate DNA barcode detection. However, the accuracy for barcode sequence discrimination was only 32%, surprisingly lower than the reported >90% accuracy for the same setup. Although the Neural Network study indeed increases the 80/20 accuracy to 80%, it remains difficult to accurately discriminate a large scale of barcode sequences. This was why the authors finally selected only three barcode sequences containing different number of abases (0, 1, and 2) to validate the multiple DSD detection. But the nanopore has already been shown to detect the numbers of abase in a DNA sequence with much higher sensitivity and accuracy (e.g., Nat Nanotechnol 2010,5:798-806).

1.2 We appreciate the reviewer's concerns with accuracy, but we are afraid the reviewer is conflating reading barcodes by *sequencing* (which is what the reviewer appears to be referencing) vs reading barcodes by *sensing* (which is what we are demonstrating here). Although sequencing-based readout has the advantages of greater barcode space and potentially higher single-molecule identification accuracies, it necessitates considerable sample preparation steps, such as an enzymatic ligation step to attach the adaptor molecules to the DNA strands. These additional steps allow for a helicase motor protein to progressively ratchet the DNA through the pore, thereby enabling sequencing-based readout. We did not employ this approach because the adaptor preparations steps would not be compatible with reading the real-time kinetics of the output strand concentration over time – this is an important feature of our enzyme-free reporter method as it requires no additional sample preparation steps. As we demonstrate in this paper, our real-time reporter strategy is more directly comparable to fluorescence-based barcoding (as opposed to sequencing-based barcoding).

...In addition, the attachment of streptavidin to a DNA used in this report has already been extensively applied in detecting DNA sequence alteration, including nucleotide substitution and base modifications (PNAS 2009, 106, 7702-7707; PNAS 2012 109, 11504-11509; J Am Chem Soc 2010, 132, 17992–17995).

1.3 The novelty in our work is that we've coupled this strep-based immobilization strategy to the upstream activity of a DSD reaction to achieve unparalleled nanopore-sensing based DSD multiplexing. The references the reviewer cites were mainly focused on exploring the sensitivity of nanopore sensors, as opposed to demonstrating a practical application for the strategy. We have added citations for these works on Page 3, Lines 93-96.

"...when conjugated to streptavidin, this modification would prevent displaced ssDNA from fully translocating through the nanopore after its free 5' end is electrophoretically captured. The nanopore ionic current signal would then be dependent on the sequence of the strand immobilized within the pore [PNAS 2009, 106, 7702-7707; PNAS 2012 109, 11504-11509; J Am Chem Soc 2010, 132, 17992–17995]."

Another problem is DSD-based microRNA detections. Based on the experimental results, this DSD method does not demonstrate any superior or unique advantages over previously reported nanopore microRNA detection, as to the scale of the multiplex detection, low detection limit (sensitivity), detection of sequence-similar microRNAs and sensing in biofluid samples.

1.4 The microRNA detection experiment was performed to show the applicability of our strategy to more clinically relevant RNA sequence inputs rather than showcase the full extent of our multiplexing potential. With regard to the scale of our multiplexed detection abilities, we show in **Fig. 4** that we are able to multiplex 10 different DSD circuit barcodes in a single experiment, with the potential to scale to 36 barcodes at the cost of slightly lower classification accuracy. No other real-time/kinetic reporting strategies that we are aware of, nanopore-based or otherwise, come close to this level of multiplexing.

1. What is the function of the fuel strand in the catalytic DSD reaction? Is it to accelerate the DSD speed or to increase the overall displacement percentage up to 100%? The work should report the kinetics without the fuel strand. If the fuel strand is functional, it should be clarified why the fuel strand was not used to catalyze the DSD circuit for any microRNA detection? Or is the fuel strand has negative effect on the microRNA detection?

1.5 The fuel strand allows the input strand to be reused, thus increasing the overall displacement percentage up to 100% [Qian Science 2011]. We have now performed an additional kinetics experiment using a plate reader to compare the difference between a catalytic DSD reaction and a non-catalytic reaction (**Supplementary Fig. S1**). The experiment shows that the output of a catalytic reaction (i.e., with a fuel strand) indeed drove the overall displacement percentage up to 100% while a non-catalytic reaction did not. The manuscript is modified as below to make this clearer (Page 3, Lines 105-112).

*“Seesaw circuits have previously been used to build large-scale logic circuits and neural networks^{2,4,9}. In this circuit architecture (Circuit 1 with Barcode A1, see **Supplementary Tables S1 and S2**), an input strand displaces a gate-bound output strand. A fuel strand then binds to the gate and displaces the input, freeing it to trigger more of the output. As such, a seesaw gate can catalytically amplify its input, a critical step for restoring signals. A catalytic reaction can increase the overall displacement percentage up to 100%^{2,35}, and the comparison between a catalytic reaction and a non-catalytic reaction is shown in **Supplementary Fig. S1**.”*

An SNV input can still trigger a DSD circuit though it is less efficient than a correct one. If a fuel strand is present in the reaction, the reaction will be catalytic, and the displacement percentage of a SNV target will be 100% [Xi Chen, JACS. 2012]. Therefore, we did not use a fuel strand for microRNA detection so that the final triggering level (steady state) is clearly different between a SNV target and a correct one. The modified manuscript is cited below for convenience (Page 5, Lines 213-217).

“We chose three barcodes from our designed set and assigned a different one to each variants’ probe. If a particular miRNA variant is present, its corresponding probe should be triggered, allowing its barcoded output strand to be detected by the nanopore sensor array. To avoid amplifying crosstalk leakage, we did not use fuel strands for our probes.”

2. Fig.2a and the sequences in the tables show that after the first DSD, the duplex domain T*DS* in the gate complex is changed to DS*T* in the intermediate. The two duplexes are identical in the melting temperature, GC content, and length (19 bps), and free energy. So, can the released output re-bind the toehold on the gate strand (left T*) to reversely displace the input, if the displacement is driven by energetic favorite? Again, the kinetics to get the equilibrium displacement percentage without the fuel strand should be detected.

1.6 With regard to binding energy to the gate, there is no difference between the input strand and the output strand. Adding the input strand to the gate will trigger some output strands from the gate to reach a new equilibrium. Also, the output strand released from a gate will irreversibly bind to a reporter complex due to the energy gain from the toehold binding. In **Supplementary Fig. S1**, we show the reaction kinetics of the seesaw gate when the fuel is not present.

3. For the measurement of the DSD kinetics, the saturate frequency of the output strand has been used to normalize the output concentration. However, this parameter only shows the time-dependent change but cannot show the actual DSD conversion efficiency (i.e. the percentage of the free output when getting the equilibrium. To calculate the DSD efficiency, it is strongly suggested that the frequency of free output strand alone (without any gate, input, and fuel strands) be used as the maximal frequency in place of the saturate frequency to normalize the output concentration.

1.7 We have now revised the nanopore data in **Figure 2c** such that the capture frequency of the free output strand, as measured on its own in a separate sample, is used as the maximal frequency for normalization. The conversion efficiency in this experiment is nearly 100% due to the inclusion of fuel strands. We have also updated Page 7-8, Lines 307-314 in the **Methods** to describe this normalization strategy.

“The average time between captures (TBC) was calculated for each five-minute interval throughout the run and normalized as follows, where $TBC_{background}$ is the average TBC of the first five data points in the No Input sample and $TBC_{saturated}$ is the average TBC of 0.5 μ M free output strand measured on its own in a separate sample.”

$$TBC_{normalized} = \frac{TBC - TBC_{background}}{TBC_{saturated} - TBC_{background}}$$

4. It is very strange that in the mixture of various of DSD strands (gate, output, input, intermediate, fuel, the gate complex etc), only the output can be captured by the nanopore. It is also surprising that the gate complex with a 15-nt overhang cannot be captured by the nanopore, according to the report. Are these events not shown or not used to participate in data analysis? The authors may contact Oxford Nanopore to confirm these observations. It should be clarified under what conditions, the duplex with an overhang can or cannot be captured by the nanopore.

1.8 Non-biotinylated nucleic acid strands do not conjugate to streptavidin and thus pass through the pore too quickly for meaningful blockades to be extracted by our analysis pipeline. This is why we do not report observed captures for the input strand, fuel strand, and intermediate complex (**Supplementary Fig. S4 and S5**).

As for the gate complex (which contains the biotinylated output strand), we have now conducted additional nanopore experiments and found that capture of the gate complex is indeed detected at roughly half the frequency compared to capture of the free output strand at the same concentration (**Supplementary Fig. S6**). Following up on this, we also found that removing the 15-nt overhang (e.g. D6 domain in **Fig. 2a**) on the same gate complex dropped its nanopore capture frequency significantly (**Supplementary Fig. S6**). The original purpose of the 15-nt overhang domain was to interact with the fluorescent reporter complex during fluorescent detection, which made a direct comparison between the two different readout methods more convenient for our work. It is, however, not required for our nanopore detection strategy alone.

We have added the following clarification on Pages 3-4, Lines 132-140.

*“We also confirmed that the circuit input strands, fuel strands, and intermediate complexes are not extracted as captures by our analysis pipeline, and thus do not contribute to our measured output concentration (**Supplementary Fig. S4 and S5**). We additionally note that the gate complex contained a 15-nt overhang to make it compatible with the fluorescence reporter gate, and that this overhang increases the background capture rate of the output strand in the pore, but is accounted for by normalizing against background (i.e. no input conditions) (see **Methods**). Capture of the gate complex can be substantially reduced by removing the D6 overhang domain from the output strand shown in **Fig. 2a (Supplementary Fig. S6)**.”*

5. The report does not show any original nanopore traces that support each analysis result. The conclusion for each figure should be associated with original traces. In addition, Figures 2, 3 and 4 were for different experiments, but they repeatedly used the same nanopore recordings.

1.9 Original nanopore raw traces for each experiment have now been added in **Supplementary Fig. S2, S5, S11, S15, and S16**. The example nanopore traces in **Figures 2, 3, and 4** have been replaced with traces from the corresponding experiments. Original nanopore raw traces for every barcode have been added in **Supplementary Fig. S8**.

Reviewer #2:

...This is a novel aspect in a field of intense research. Still some issues need to be clarified, as listed below, before the manuscript can be accepted for publication in Nature Communications.

1. Fig.2c displays a comparison of using a nanopore and a spectrofluorometer. It would be insightful to discuss shortly the discrepancy in the kinetics in the 'no input' cases.

2.1 Circuit leakage is more pronounced in the spectrofluorometer sample compared to the nanopore sample due to the interaction of the circuit gate complex with the fluorescent reporter gate complex, as described on Page 3, Lines 124-132:

*“We also noted that samples with no input added showed higher levels of output strand “leak” when using the fluorescent reporting method. We hypothesized that this circuit leakage was caused by the interaction between the seesaw gate and the fluorescent reporter gate² (**Fig. 2b**). Because our nanopore readout strategy detects the ssDNA output strand directly, it does not require an additional reporter gate, thus the reporter leak was not observed on the nanopore-based kinetics plot. To verify this hypothesis, we tested a “clamped” seesaw gate (**Supplementary Table S3**) that has previously been shown to suppress reporter leakage². Indeed, the no input sample kinetics using a clamped seesaw gate more closely matched the no input nanopore readout (**Supplementary Fig. S3**).”*

2. Could nanopore detection overcome fluorescent reporter-based detection in terms of accuracy?

The authors claim that multiplexed detection is more efficient using nanopores. However, the discussion and the results shown in Fig.5 do not strongly support this claim. If indeed multiplexed detection is superior using nanopores, a stronger focus on this should be given in the manuscript.

2.2 In terms of multiplexing, the accuracy of fluorescent reporter-based detection is limited by spectral crosstalk. In practice, no more than 4 fluorophores can typically be multiplexed together without significant crosstalk (*J. Phys. Chem. Lett.* 2018, 9, 15, 4379–4384). In contrast, we now present results demonstrating that our nanopore barcode strategy can achieve high accuracy (>90% single molecule) with 10-plex detection (Figure 3), and can scale up to 36-plex detection while still maintaining ~70% single molecule accuracy using a more powerful CNN classifier (ResNet-18) compared to our original manuscript. Further, we show that even higher accuracy classification results can be achieved taking into account the barcode classifier's output score as a measure of accuracy confidence, and only taking into account classifications that pass a preset confidence filter. For example, we found that implementing a confidence filter of 90% significantly boosted accuracy on the 36-way barcode classification test dataset from ~70% to ~93% (**Supplementary Fig. S13a**), while still maintaining a significant proportion of the data sets (**Supplementary Fig. S13b**).

We have added details of these improvements here (Page 5, lines 185-192):

"We additionally benchmarked barcode classification accuracies using ResNet-18, a more sophisticated CNN popular in computer vision applications³⁷. The ResNet-18 model yielded an improved 36-way classification accuracies of ~93% and ~67%, with and without the use of a classifier confidence filter (Fig. 4a, Supplementary Fig. S13 and Methods). From this collection, we identified a subset of ten barcodes with the most separable signal levels (Fig. 4b and Supplementary Fig. S14) and achieved an average single-molecule classification accuracy of ~96-97% after training and testing the ResNet-18 CNN on this limited set (Fig. 4c and Supplementary Fig. S13)."

The microRNA experiment in **Fig. 5** was designed to show the capability of DSD circuit specificity through SNV detection, while also demonstrating a relatively modest multiplexed diagnostic application of our reporting strategy. It was successful in terms of showing that nanopore allowed for multiplexed detection, while the relative activation level of detection probes in nanopore-based readout experiments is also consistent with that of single-plex spectrofluorometer-based experiments (**Supplementary Fig. S17**). That is, the crosstalk we observe between our let-7a and let-7c detection probes results from the DNA DSD circuit design and the challenge that SNVs pose to the specificity of strand hybridization, and is not caused by our nanopore reporting method.

3. Overall, the prediction discussed is not very high, especially compared with the accuracy of nanopores in detecting DNA.

2.3 We would like to again point out the differences in comparing accuracies derived from nanopore *sequencing* vs single-molecule *sensing*, which we explain in our previous response 1.2 above. And also note the improved accuracies we now present in response 2.2 above.

The random barcodes yielded a single-molecule classification accuracy of about 72%. Did the authors attempt to optimise their network in order to enhance the accuracy or is this the highest accuracy obtainable? Is there a way to optimise the experimental setup and conditions instead of the Machine Learning part, in order to achieve a better prediction? Is there overall room for improvement or is the accuracy of approximately 70% the upper bound? In that case, how would this compromise the use of nanopores as suggested here?

2.4 We do not think the 70% accuracy is the upper bound (see point 2.2 above). On the experimental setup side of things, we think that exploring more barcodes that include chemical modifications to the DNA is the surest way to develop expanded sets of barcodes, beyond the 36 we show here, whilst maintaining sufficient discrimination accuracy. One advantage of using our nanopore sensing strategy is that detection is not limited to native nucleotides; any chemical modification that is captured in the barcode will manifest an electrical signature. We discuss this briefly in the Discussion section of the paper.

4. From the supporting information, it can be inferred that there was no development in the Machine Learning (ML) part. Were the tools used as is? Was the feature set tested, i.e. the type of features used? It has been shown that different feature sets in the nanopore signals can have a strong impact on the detection accuracy. In case, the authors did not consider this, they should at least comment on whether there is room for optimising the ML part, type of network, features, etc. in order to enhance the detection accuracy. That would further strengthen the use of nanopores over spectrofluorometers.

2.5 We agree with the reviewer that exploring classification models and parameters is an area that could potentially lead to improved results. In our initial manuscript, we explored two different types of classification models: a relatively “basic” 4 layer CNN and Logistic Regression (as a simple baseline). In response to this reviewer suggestion, we have now added additional model analyses, including the use of a Random Forest model, and a more powerful CNN (ResNet-18).

We trained the Random Forest model using the mean, median, standard deviation, maximum, and minimum signal features from our raw captures. We characterized the importances of these features to learn how much they contributed towards classifier prediction accuracy (**Supplementary Fig. S10**). We did not try additional features beyond these five, so there is likely room for improvement. However, we ultimately decided to switch to CNN models because they do not rely on a limited, predetermined set of signal features, and they yielded higher prediction accuracies on average. We have added a brief description of our Random Forest model on Page 4, Lines 158-160:

*“Meanwhile, a Random Forest model trained on five features of the raw signal (mean, median, standard deviation, maximum, and minimum) yielded an accuracy of ~64% (**Supplementary Fig. S10**).”*

Following the reviewer’s suggestion, we also continued to explore additional ML models and

found that a more powerful CNN architecture (ResNet-18 CNN), which is commonly used in Computer Vision applications, resulted in increased classification accuracies. We discuss the use of the ResNet18 classifier in response 2.2 above.

5. Regarding the experimental setup and conditions: Were other conditions, such as the molecules concentration etc, also tested? Were other circuits also used? If so, were the monitoring in those cases also successful?

2.6 In the experiments presented in this paper, we ran seesaw catalytic circuits at a gate complex concentration of 0.5 μM (**Fig. 2c** and **Fig. 3e**) and 0.2 μM (**Fig. 4d**) on the nanopore with all cases being successful. Based on our standard curve from **Fig. 1d**, we expect our detection strategy to be sensitive to output strand concentrations at least as low as 20 nM. We propose strategies to increase sensitivity as future work in the Discussion.

We tested our detection strategy with two types of circuits in this paper: seesaw catalytic amplifiers (**Fig. 2a**) and competitive three-stranded probes (**Fig. 5a**). Both circuits showed similar reaction kinetics as compared using fluorescent-based reporters, indicating that our nanopore detection strategy can be a generalizable approach to reading DSD reactions. Interestingly, we also now performed additional experiments and observed that seesaw gates with overhangs showed higher signal background because gates with overhang can be read by nanopore readers. Indeed, we demonstrated that removing the overhangs can effectively reduce such background. While only two types of circuits were tested, the result can be generalized to other DNA gates because most DNA gates have similar multi-stranded gate structures with or without overhangs.

6. The authors attempted to focus on the advantages of nanopores over spectrofluorometers. Are there, though, any limitations in the use of nanopores for a DSD kinetics readout? How barcode-specific is the framework proposed here?

2.7 One potential limitation of our current detection strategy is its sensitivity. As shown in our standard curve (**Fig. 1d**), the lowest concentration for which we characterized the output strand was 20 nM. At lower concentrations, samples need to be analyzed for longer durations in the nanopore flow cell to obtain enough captures for confident analysis, as our quantification strategy relies on averaging the capture rate over many single-molecule events.

In addition, although our detection strategy is useful for multiplexing different circuits within the same sample, it is currently not possible to multiplex samples of differing experimental conditions (e.g. buffer concentration, temperature) within the same flow cell. In the latter case, multiple flow cells would need to be run in tandem, an endeavor that can eventually be made easier with arrays of spatially-isolated nanopore sensors, such as Oxford Nanopore's 96-well nanopore device, the Plongle (<https://nanoporetech.com/products/plongle>).

These limitations have been summarized and added to Page 6, Lines 232-235.

“However, some current limitations include a lower sensitivity compared to fluorometers, lower time resolution, and the inability to multiplex samples under different conditions due to the lack of physical barriers to separate samples in a flow cell.”

7. How prone are the analysed results with respect to the filter that checks whether five of its signal features (mean, median, minimum, maximum, standard deviation) are within the expected range? Were the filters and the classifier optimised in advance to this study?

2.8 We tuned our filter parameters empirically. Specifically, for every different barcode we tested, we plotted raw current distributions for each filter parameter (e.g. raw current distribution of signal mean in **Figure 1c**) and designed our parameter values to encapsulate each barcodes' signal feature peak distribution. These details have now also been added to the **Methods** section on Page 9, Lines 365-368.

8. The training was performed with 3000 samples. The data set is not very large. Did the authors check, whether their results on the accuracy do converge or are prone to the sample size?

2.9 We have now evaluated the impact of training sample size on classifier accuracy for three different types of classifiers we used in this work. We observed that classifier accuracy plateaus at 2000 examples per class for the Random Forest and ResNet-18 classifiers, and at 7000 examples per class for the simpler CNN classifier (see new **Supplementary Fig. S19**). We have added a brief reference to this supplementary information on Page 6, Lines 235-237.

*“Another area for improvement is classifier prediction accuracy, which can be influenced by classifier type and number of training examples (**Supplementary Fig. S19**), or by imposing a confidence threshold (**Supplementary Fig. S13**).”*

Minor Comments:

1. How is the steady state defined? In other words, is it certain that the steady state has been reached?

2.10 When performing circuit endpoint measurements on the nanopore, we first measure the time it takes for the same circuits to reach steady state on a spectrofluorometer and ensure they are given the same amount of time to reach steady state prior to analysis on the nanopore. This clarification has been added to **Methods** on Page 8, Lines 326-332.

“Samples for the five-circuit multiplexing experiment consisted of 0.2 uM gate complex from each circuit and a total of 3.2 uM streptavidin suspended in 1X C17 to a total volume of 200 uL. 0.4 uM of each desired input was added to the samples, which were then immediately placed in a 30 °C incubator and allowed to react to steady state over the course of three hours (circuits were previously characterized on a fluorometer to ensure three hours is adequate time to reach steady state, which we define as the point when output fluorescence does not change over 5% for at least one hour).”

2. Typesetting error: on pg.9, line 5, do the authors mean 'All feature classification of capture events' instead of 'All future classification of capture events'?

2.11 We have changed the wording on Page 9, Line 376 for clarity.

“All subsequent classification of capture events were performed by Convolutional Neural Networks (CNNs) via PyTorch.”

3. Shortly discussing in the end the impact of the presented results and their applicability would add to the manuscript.

2.12 We have added additional discussion content to our manuscript on Page 6, Lines 229-248.

*“In summary, we have developed a new reporter strategy for DSD reactions using nanopore sensing. Our system holds key advantages over fluorescence-based methods, including greater multiplexing and real-time readout using an inexpensive, portable device with flow cells that can be re-used for multiple analytical samples (**Supplementary Fig. S18**). Further, compared to fluorophore-quencher pair-based reporter systems, our method uses DNA sequence-based barcodes (not small molecules) and so only requires a single type of DNA modification (biotinylation), which is simpler to synthesize and is not susceptible to photobleaching. However, possible limitations of our nanopore interface include a lower sensitivity compared to fluorimeters, lower time resolution, and the inability to multiplex samples under different conditions due to the lack of physical barriers to separate samples in a flow cell. Another area for potential improvement is classifier prediction accuracy, which can be influenced by classifier type and number of training examples (**Supplementary Fig. S19**), or by imposing a confidence threshold (**Supplementary Fig. S13**). Future work will be aimed at: 1) further expansion of the barcode space, for example, with chemical modifications to the DNA that could expand the dynamic range of barcode signal space⁴⁰, and/or the ability to read sequential barcode regions within a single output strand^{25,41,42}; and 2) increasing the reaction speed and sensitivity of DSD reactions, for example by spatially-localizing DNA components to the nanopore sensor membrane^{43,44}. Increasing the scale and speed of our detection strategy for more complex molecular computing architectures, such as cascaded circuits or oscillators, will further take advantage of our method’s ability to generate both multiplexed and kinetic readouts. These advancements would expand the capabilities of molecular computing tools by facilitating the design and facile characterization of more complex circuits, bringing forward new opportunities for their application in medical diagnostics⁴⁵, therapeutics⁴⁶, biomolecular-based instruments⁴⁷ and molecular information processing^{4,8}.”*

Reviewers' Comments:

Reviewer #1:

Remarks to the Author:

Considering the effort in experimental improvement, this reviewer agrees the publication of The revised manuscript.

Reviewer #2:

Remarks to the Author:

Referee report

Re: NCOMMS-21-45691A

A nanopore interface for high bandwidth DNA computing

by Karen Zhang, Yuan-Jyue Chen, Kathryn Doroschak, Karin Strauss, Luis Ceze, Georg Seelig, and Jeff Nivala

The authors have provided a revised version of their work entitled as given above. They have replied to all referee comments and have respectively added comments in their manuscript. In this way, the authors have improved their manuscript, addressing unclear and controversial points. In its current form, this work can be accepted for publication in Nature Communications.